# Peptide inhibition of neutrophil-mediated injury after *in vivo* challenge with supernatant of Pseudomonas aeruginosa and immune-complexes

**Adrianne Enos[1], Parvathi Kumar[1,2]\*, Brittany Lassiter[1], Alana Sampson[1], Pamela Hair[1], Neel Krishna[1,3,4], Kenji Cunnion[1,2,3,4,5]**

**1** ReAlta Life Sciences Inc, Norfolk, Virginia, United States of America, **2** Children's Hospital of The King's Daughters, Norfolk, Virginia, United States of America, **3** Department of Microbiology and Molecular Cell Biology, Eastern Virginia Medical School, Norfolk, Virginia, United States of America, **4** Department of Pediatrics, Eastern Virginia Medical School, Norfolk, Virginia, United States of America, **5** Children's Specialty Group, Norfolk, Virginia, United States of America

\* pkumar@realtals.com

**Data Availability Statement:** All relevant data are within the manuscript and its Supporting information files.

## Abstract

Neutrophils are recognized for their role in host defense against pathogens as well as inflammatory conditions mediated through many mechanisms including neutrophil extracellular trap (NET) formation and generation of reactive oxygen species (ROS). NETs are increasingly appreciated as a major contributor in autoimmune and inflammatory diseases such as cystic fibrosis. Myeloperoxidase (MPO), a key neutrophil granule enzyme mediates generation of hypochlorous acid which, when extracellular, can cause host tissue damage. To better understand the role played by neutrophils in inflammatory diseases, we measured and modulated myeloperoxidase activity and NETs *in vivo*, utilizing a rat peritonitis model. RLS-0071 is a 15 amino acid peptide that has been shown to inhibit myeloperoxidase activity and NET formation *in vitro*. The rat model of inflammatory peritonitis was induced with intraperitoneal injection of either *P. aeruginosa* supernatant or immune-complexes. After euthanasia, a peritoneal wash was performed and measured for myeloperoxidase activity and free DNA as a surrogate for measurement of NETs. *P. aeruginosa* supernatant caused a 2-fold increase in MPO activity and free DNA when injected IP. Immune-complexes injected IP increased myeloperoxidase activity and free DNA 2-fold. RLS-0071 injection decreased myeloperoxidase activity and NETs in the peritoneal fluid generally to baseline levels in the presence of *P. aeruginosa* supernatant or immune-complexes. Taken together, RLS-0071 demonstrated the ability to inhibit myeloperoxidase activity and NET formation *in vivo* when initiated by different inflammatory stimuli including shed or secreted bacterial constituents as well as immune-complexes.

**Funding:** This work was funded in part by NIH grant R21 AI135222. The funder, ReAlta Life Sciences Inc., provided support in the form of salaries for authors [AE, PSK, BL, AS, PSH, NKK, KMC], but did not have any additional role in the study design, data collection and analysis, decision to publish, or preparation of the manuscript. The specific roles of these authors are articulated in the 'author contributions' section.

**Competing interests:** Parvathi Kumar, Adrianne Enos, Brittany Lassiter, and Pamela S. Hair: employees of ReAlta Life Sciences. Neel K. Krishna is an employee and has equity ownership of ReAlta Life Sciences Kenji. M. Cunnion is an employee, has equity ownership, and membership on the board of ReAlta Life Sciences. Commercial affiliation with ReAlta Life Sciences Inc, does not alter our adherence to PLOS ONE policies on sharing data and materials.

# Introduction

Cystic fibrosis (CF) is the most common, frequently fatal inherited disease in the Caucasian population [1]. It is a chronic and progressive disease caused by defective or absent cystic fibrosis transmembrane conductance regulator (CFTR) chloride channels in the body affecting the pulmonary and gastrointestinal systems predominantly. The 2018 CF foundation (CFF) patient registry reports > 30,000 individuals are living with CF in the United States of America (USA) [2]. Chronic pulmonary obstruction leading to respiratory failure is the primary cause of death in > 90% of individuals with CF [3]. In their airways the dysfunction or absence of CFTR channels results in accumulation of abnormally viscous mucous which interferes with mucociliary clearance of pathogenic bacteria. Most notable of these bacterial pathogens *Pseudomonas aeruginosa*, *Staphylococcus aureus* and *Burkholderia cepacia* then infect the airways causing a chronic inflammatory environment propagated by the failure of microbial clearance creating a pro inflammatory and toxic microenvironment [4]. Progressive destruction of lung parenchyma is mediated by a cycle of recurrent infections, neutrophil dominated inflammation, and airway obstruction with impaired mucociliary clearance [3,5].

Neutrophils, the first cells to migrate into the lung tissue are recruited by powerful anaphylatoxins including C5a to target bacterial and fungal pathogens, but their activation also results in damage to the surrounding tissue due to release of oxidants and proteases locally [6]. Further neutrophils are the main source of viscous DNA which contributes to obstruction of airways [7]. As the cycle repeats there is progression from lung damage to lung scarring and finally pulmonary failure [8].

Neutrophil extracellular traps (NETs) arise from neutrophils extracellularly and are composed of chromatin fibers decorated with cytoplasmic proteins such as neutrophil elastase and MPO as well as histone. Many studies have now clarified the role played by neutrophil cytotoxins, extracellular DNA and NETs in CF associated lung injury [9–14]. Unregulated and excessive NET formation in pulmonary alveoli are detrimental to the lung resulting in NET-mediated tissue damage [15].

An additional contributor to parenchymal lung damage in CF is extracellular MPO, another enzyme elaborated from neutrophil granules or on NETs [16–18]. Myeloperoxidase (MPO) is a heme-based peroxidase present in neutrophil granules and is a major constituent of the dry weight of neutrophils [19,20]. It is a major contributor to oxidative inflammation in a wide range of human inflammatory diseases including cystic fibrosis, chronic obstructive pulmonary disease, systemic lupus erythematosus, and autoimmune kidney diseases [16,21–23]. MPO catalyzes the generation of hypochlorous acid (HOCl) from hydrogen peroxide and chloride ion [24,25]. In CF, neutrophils recruited into the lungs [26] in response to infection with pathogenic bacteria, like *Pseudomonas aeruginosa (P. aeruginosa)*, trigger the release of MPO catalyzing the generation of HOCl, which when extracellular, damages the delicate lung tissue [27].

The complement system is the most destructive inflammatory cascade in the human body and two of the four most prevalent proteins in the CF lung fluid are complement effector molecules, C3 and C4 [28]. Complement activation yields robust pro-inflammatory complement effectors including C5a which are associated with lung disease severity in CF, for example C5a concentration in the sputum from patients with CF has been shown to be inversely correlated with the body mass index (BMI) [29,30] which is often associated with worse pulmonary function [31,32]. Immune complexes (ICs) are potent initiators of classical pathway complement activation generating robust proinflammatory complement effectors such as the anaphylatoxin C5a, which recruits and activates neutrophils, inducing vascular leakage and

bronchoconstriction [33]. ICs are known to be integral in the pathogenesis of CF [34] as well as other autoimmune conditions like systemic lupus erythematosus and immune vasculitis.

RLS-0071, previously reported as Peptide Inhibitor of Complement C1 (PIC1), is a 15 amino acid PEGylated synthetic peptide shown to inhibit MPO activity in sputum collected from patients with CF [35] and inhibit the oxidase activity of other heme-based enzymes like hemoglobin and myoglobin [36]. RLS-0071 can potently inhibit the peroxidase activity of MPO *in vitro*, preventing the oxidation of target molecules [37]. The antioxidant activity is mediated via single electron transport and hydrogen atom transfer [38]. In addition to RLS-0071's ability to inhibit antibody-initiated complement activation, we have also demonstrated that RLS-0071 can inhibit MPO-mediated PMA-initiated (Phorbol 12-myristate 13-acetate) NET formation by human neutrophils [39]. We speculate that RLS-0071 could potentially exert three positive effects on CF lung disease: block MPO-mediated generation of HOCl, block the generation of NETs and block the generation of complement effectors.

To measure RLS-0071's inhibitory effect on MPO and NETosis *in vivo*, we evaluated a rat inflammatory peritonitis model mediated by intraperitoneal injection of purified human MPO, *P. aeruginosa* supernatant, or immune-complexes [40]. The peritoneum has been used in other animal models to evaluate inflammatory responses, such as reverse passive arthus [41] and immune response to infection [42].

## Materials and methods

### Ethics statement and animal welfare

Animal research was conducted under approval from the Eastern Virginia Medical School Institutional Animal Care and Use Committee (IACUC # 17–008) and was in accordance with the National Institutes of Health's Guide for the Care and Use of Laboratory Animals. In keeping with current ethical standards of animal use the numbers of rats utilized were the minimal needed to achieve meaningful data [43]. Adolescent male Wistar rats were purchased from Hilltop Lab Animals (Scottdale, PA, USA). All procedures were performed under anesthesia while keeping the animals sedated and comfortable. Animals were monitored throughout the study duration by laboratory personnel and the following humane endpoints applied to all animals. If the animal reached humane endpoints, it was euthanized. The animals' welfare was assessed by observing the following signs: general appearance (dehydration, weight loss, abnormal posture, condition of skin and fur, signs of pain); ambulation (reluctance or difficulties to move); behavior (apathy, abnormal behavior); clinical signs (eating, drinking, urinating, defecating). If there was a deviation from normal, the animal was closely monitored and treated, when possible (e.g., hydration, analgesia, warming). As a general rule, if the animal was monitored longer than 12 hours and if its condition had not markedly improved, the animal was euthanized. While under sedation, the rats were kept on a heating pad, visually monitored for respiratory effort, behavior and monitored every 15 minutes using a pulse oximeter to check heart rate and oxygen saturation. Animals were deeply anesthetized with a cocktail of ketamine/acepromazine to collect blood for separation of plasma and subsequently euthanatized using FatalPlus (Patterson Vet Greeley, CO) to collect peritoneal fluid.

### Reagents

RLS-0071 (IALILEPICCQERAA-dPEG24) was manufactured by PolyPeptide Group (San Diego, CA) to $\geq$ 95% purity verified by HPLC and mass spectrometry analysis. Lyophilized RLS-0071 was solubilized in a 0.05 M Histidine buffer pH 6.7 (solubilized in normal saline with 0.01 M $Na_2HPO_4$ buffer to 37.5 mM). Purified myeloperoxidase (MPO) was purchased from Lee Bio solutions (Maryland Heights, MO). Commercial ready-to-use

tetramethylbenzidine (TMB) was purchased from Thermo Fisher (Waltham MA). This 1-Step™ Ultra TMB-ELISA Substrate Solution uses 3,3',5,5'-Tetramethylbenzidine (TMB) and detects horseradish peroxidase (HRP) activity, yielding a blue color that changes to yellow upon addition of a sulfuric or phosphoric acid stop solution. PicoGreen were purchased from Thermo Fisher (Waltham MA) and ovalbumin from Sigma (St. Louis, MO).

## Animal experiments

Male Wistar rats approximately 6–9 weeks old, 200–250 grams in weight were used. For all procedures, rats were sedated with ketamine (McKesson, Las Colinas, TX, USA) and acepromazine (Patterson Veterinary, Saint Paul, MN, USA) throughout the course of the experiment. Rectal temperature and respiratory efforts were monitored between sedation and euthanasia was completed 4 hours later. Blood draws are obtained from the tail vein while under sedation.

**Intraperitoneal injections of thioglycolate broth.** Injection of thioglycolate broth (Sigma Aldrich St. Louis MO) into the peritoneal cavity 24 hours prior to the experiment elicits a robust influx of neutrophils into the peritoneal cavity [44]. Rats were injected IP with 2ml of 4% thioglycolate broth, prior to the animal experiments as described below.

**Preparation of *P. aeruginosa* supernatant.** Pseudomonas stationary broth was prepared by growing *P. aeruginosa* (ATCC, Manassas, Virginia) in Mueller Hinton broth (Difco BD, Franklin Lakes, NJ) in a rotating shaker at 37°C overnight to establish a stationary phase culture. This culture was spun at high speed and the supernatant was removed from the pellet. The *P. aeruginosa* supernatant (Ps.a supe) was then sterile filtered through a 0.2 um filter unit to ensure removal of any bacteria. This Ps.a supe rich in secreted exotoxins and extracellular inflammatory metabolites was then used for *in vivo* experiments.

**Preparation of immune-complexes.** ICs were made a day prior, by adding ovalbumin (Sigma, St. Louis, MO) at a final concentration of 1 mg/ml to rabbit anti-ovalbumin sera. This solution was allowed to incubate at room temperature for 30 minutes, 37°C for 30 minutes, wet ice bath for 30 minutes, and then overnight at 4°C. ICs were visible the next day and washed three times with PBS before IP injection into the rats.

**Rat peritoneal fluid.** Peritoneal fluid was obtained from the peritoneal cavity after IP infusion of 20ml of ice-cold saline solution. This effluent was spun down to separate out the cells and the fluid was used to perform TMB measurement and free DNA concentration analysis. The detached cells spun down into a pellet were washed in additional saline and resuspended to their original volume to perform cell count and preparation of slides for DAPI staining.

**MPO dose ranging, and time course experiments followed by rescue with RLS-0071.** A pilot experiment was performed to establish the intraperitoneal (IP) MPO dose ranging conditions with increasing amounts of purified MPO (0.01, 0.03 and 0.1mg/ml) injected IP. Animals were allowed to wake up between blood draws and were re-sedated before the terminal blood draw. After 1 hour, the animals underwent euthanasia, and a peritoneal wash was done; 20 ml of ice-cold saline was injected into the peritoneum and the fluid was retrieved before necropsy was performed.

In subsequent experiments, rats were injected with purified MPO intraperitoneally to determine the optimal timing of MPO-mediated neutrophil activation as assessed by free DNA and MPO in the peritoneal fluid. Rats were injected with 0.1mg of purified MPO at 0 hour and then euthanized and a peritoneal lavage was performed at 1,2- and 4-hour time point.

To assess the efficacy of RLS-0071 in this inflammatory model, rats were injected 0.1mg of purified MPO intraperitoneally followed by IP injection of RLS-0071 in three increasing doses: 1 mg (4mg/kg), 5mg (20mg/kg) or 20mg (80mg/kg). After 2 hours the rats were euthanized

and peritoneal washes with 20ml of ice-cold saline were performed. The peritoneal washes were sedimented for cells and the supernatant was measured for MPO activity with a TMB-based assay [45] which is the most commonly utilized substrate for testing MPO activity. Oxidation of TMB results in formation of TMB diamine [46] and this color change can be read on a spectrophotometer [36].

**P. aeruginosa peritonitis experiments measuring MPO release and NET formation.** To generate conditions for *P. aeruginosa* supernatant induced peritonitis, rats received IP injection with thioglycolate broth 24 hours prior to experiment start. Twenty-four hours later *P. aeruginosa* supernatant (Ps.a supe) or saline was administered IP. RLS-0071 at low dose (10 mg or 2.5mg/kg) and high dose (160 mg or 40mg/kg) was injected immediately after Ps.a supe injection and euthanasia was performed 4 hours later. Peritoneal fluid was collected by ice-cold saline lavage and analyzed for neutrophil migration, MPO activity and free DNA. Blood samples were collected, and plasma purified for detection of free DNA.

**Immune-complex mediated inflammatory peritonitis.** In order to test the ability of RLS-0071 to moderate MPO activity related to inflammation induced by intraperitoneal injection of pre-formed ICs, increasing doses of RLS-0071 were injected IP immediately after IC injection and euthanasia was performed 4 hours afterwards. Rats received IP injection with thioglycolate broth 24 hours prior to experiment start and the inflammatory peritonitis was furthered by injecting the pre-formed ICs. After euthanasia, peritoneal lavage was performed using ice-cold saline lavage and analyzed for neutrophil migration, MPO activity and free DNA released. Blood samples were collected, and the plasma was analyzed for free DNA.

## Quantitation of MPO peroxidase activity

In a 96 well plate, 0.1ml of recovered peritoneal fluid was titrated down 1:2 and mixed with TMB (0.1 ml) was added to each well for 2 minutes and followed by 0.1 ml of 2.5 N $H_2SO_4$ for another 2 minutes and then read on a 96 well plate reader (BioTek) at 450 nm. Purified MPO was used for generating a standard curve.

## Quantitation of NET formation

Quantitation of free DNA released from the neutrophils was performed using PicoGreen (Molecular Probes). In a 96 well plate, 0.1ml of recovered peritoneal fluid and rat plasma was titrated down 1:2 and mixed 1:1 with prepared PicoGreen reagent. The preparation was left to incubate in the dark for 10 minutes and fluorescence was then quantified on a BioTek microplate reader at excitation 485nm/emission 528nm DNA was used to generate a standard curve.

## NET formation visualized by fluorescence microscopy

Peritoneal fluid was obtained and cytospun onto a glass slide. Cells were incubated in a blocking solution (2% normal goat serum + 2% bovine serum albumin in PBS) for 1 hour at room temperature. Slides were washed in PBS three times and incubated in DAPI (Southern Biotech) at 0.25 mg/ml final in 2% BSA in PBS for 1 hour at room temperature. Slides were then washed three times in PBS and imaged. Cells were visualized using a DP70 Digital Camera (Olympus Center, Valley Forge, PA, USA), mounted on a BX50, Olympus microscope.

## Statistical analysis

Quantitative data were analyzed determining means, SEM, and Student's *t*-test [47] using Excel (Microsoft, Redmond, WA, USA).

## Results

### RLS-0071 inhibition of MPO peroxidase activity *in vivo*

To isolate MPO oxidative activity and induction of NET formation *in vivo*, we conducted initial pilot studies with purified MPO injected IP. Dose response pilot experiments depicted in supplemental figure S1 Fig, show that a 0.1mg MPO dosed IP demonstrated a trend towards increased peroxidase activity (p = 0.12) in the peritoneal lavage fluid collected at 1 hour after dosing as measured using a TMB-based assay (S1A Fig). A major component of NET is extracellular DNA which was measured as free DNA released from neutrophils into the peritoneal fluid and plasma collected at 1 hour after dosing. S1B Fig shows the free DNA measured in the peritoneal wash fluid trended towards increased free DNA as noted for the 0.1mg MPO dose (p = 0.19). Overall, increasing amounts of MPO injected demonstrated 6 to11-fold increases in peroxidase activity compared with saline injection suggesting a dose-response relationship. The time course experiment (S1C Fig) conducted to evaluate optimal dwell time for the purified MPO in the peritoneum demonstrated a 10-fold increase in MPO activity induced 2 hours after 0.1mg of MPO was injected intraperitoneally (p = 0.005) without further increase at 4 hours. S1D Fig shows greater than 7-fold increase of free DNA measured in the peritoneum at 2 (p = 0.03) and 4 (p = 0.07) hours after 0.1mg IP MPO.

We then tested increasing doses of RLS-0071 injected IP immediately after purified MPO was inoculated into the peritoneum. A 20 mg dose (80 mg/kg) of RLS-0071 demonstrated a 5-fold decrease (p = 0.015) in peroxidase activity (Fig 1A) compared with no RLS-0071 injection after MPO IP administration. Significant decreases in MPO activity (p < 0.019) were also demonstrated for RLS-0071 doses of 5 mg (20 mg/kg) and 1 mg (4 mg/kg). A trend towards decreased free DNA in the peritoneal fluid (p = 0.11) was noted for the 20 mg (80 mg/kg) dose of RLS-0071 compared with no RLS-0071 (Fig 1B). Free DNA in plasma also demonstrated a similar trend towards a decrease in the highest dose of RLS-0071 (p = 0.22) compared with no RLS-0071 after MPO injection. These results demonstrate that RLS-0071 can decrease MPO-mediated peroxidase activity *in vivo* as well as modulate NETosis.

### RLS-0071 effects on *P. aeruginosa* supernatant (Ps.a supe) induced peritonitis

To evaluate the extent to which RLS-0071 can modulate *P. aeruginosa* induced inflammation, we adapted the inflammatory peritonitis model by injecting *P. aeruginosa* supernatant prepared from overnight growth in media containing secreted and shed bacterial components. Thioglycolate broth was injected IP to gently recruit neutrophil into the peritoneal cavity over 24 hours [44] prior to injection of inflammatory *P. aeruginosa* supernatant. Blood was obtained prior to euthanasia and peritoneal wash was performed after euthanasia. S2 Fig depicts cells counted using a hemocytometer confirmed to be neutrophils on Wright stain. Please note that the slides are concentrated cytospins meant to serve only as qualitative representative images and are not intended for quantitative measurements. Fig 2A shows a trend of 2-fold increase of intraperitoneal neutrophil recruitment induced by *P. aeruginosa* supernatant compared to saline (p = 0.07) and Fig 2B shows a 2-fold increase in MPO peroxidase activity (p = 0.04) compared to saline. Fig 2C and 2D show *P. aeruginosa* supernatant induces ≥ 2-fold increases in free DNA, as a measure of NETs, in the peritoneal fluid (p = 0.008) and plasma (p = 0.1), respectively. Fig 2E and 2F show images of normal neutrophils, without NETs, following IP saline injection compared to NETs induced by *P. aeruginosa* supernatant injected IP. Taken together this shows that *P. aeruginosa* shed and secreted products induce peritoneal inflammation characterized by neutrophil infiltration, MPO activity and NET formation.

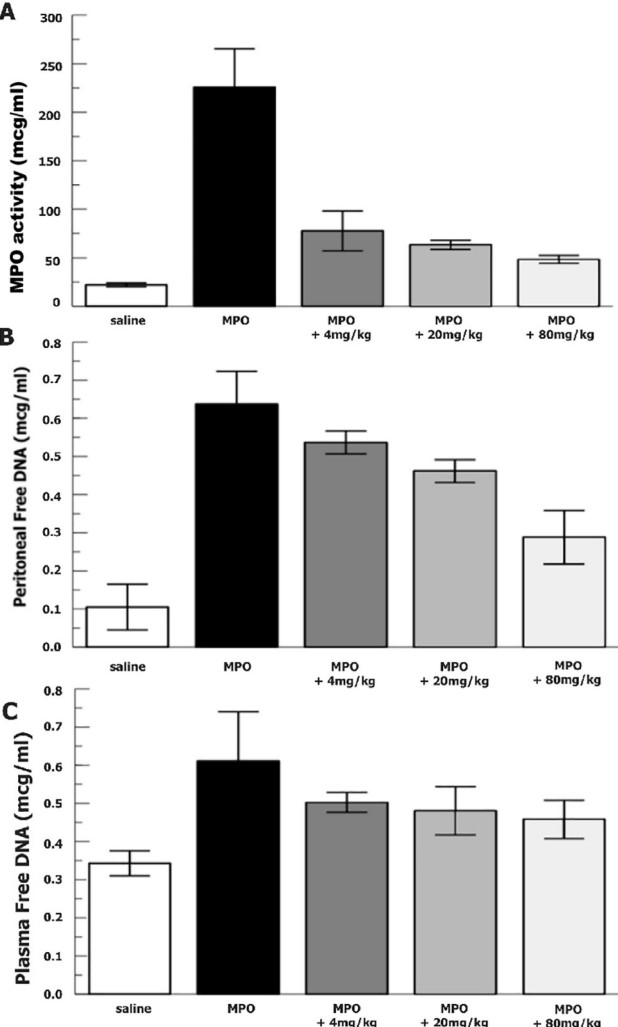

**Fig 1. Intraperitoneal MPO with increasing doses of RLS-0071.** Purified MPO (0.1 mg) was injected IP immediately followed by RLS-0071 injection IP at increasing doses. Two (2) hours after IP injections animals underwent phlebotomy, euthanasia, and peritoneal wash. A) Peritoneal wash supernatant oxidation of TMB (n = 4). Data are means of independent animals ±SEM. B) Peritoneal wash supernatant free DNA measured via PicoGreen assay (n = 4). Data are means of independent animals ±SEM. C) Blood plasma free DNA measured via PicoGreen assay (n = 4). Data are means of independent animals ±SEM.

To ascertain the effect of RLS-0071 on *P. aeruginosa* induced inflammation, two doses of the peptide were injected IP immediately after *P. aeruginosa* administration. Fig 3A demonstrates a >2-fold neutrophilia (p = 0.008) induced by *P. aeruginosa* IP supernatant compared to saline and a dose-dependent reduction in the intraperitoneal neutrophil migration with increasing doses of RLS-0071. The high dose of RLS-0071 at 160 mg/kg IP led to a 50% reduction in peritoneal neutrophils compared to *P. aeruginosa* alone relative to the saline baseline (p = 0.05). Fig 3B shows a 2-fold decrease in MPO activity with high dose RLS-0071 at 160 mg/kg compared to MPO activity induced by *P. aeruginosa* supernatant alone (p = 0.005). Fig 3C and 3D demonstrates dose dependent RLS-0071 inhibition of *P. aeruginosa* supernatant induced NET formation as measured by free DNA released in the peritoneal fluid and plasma. Fig 4 show representative images of stained slides of peritoneal wash supernatants. Please note

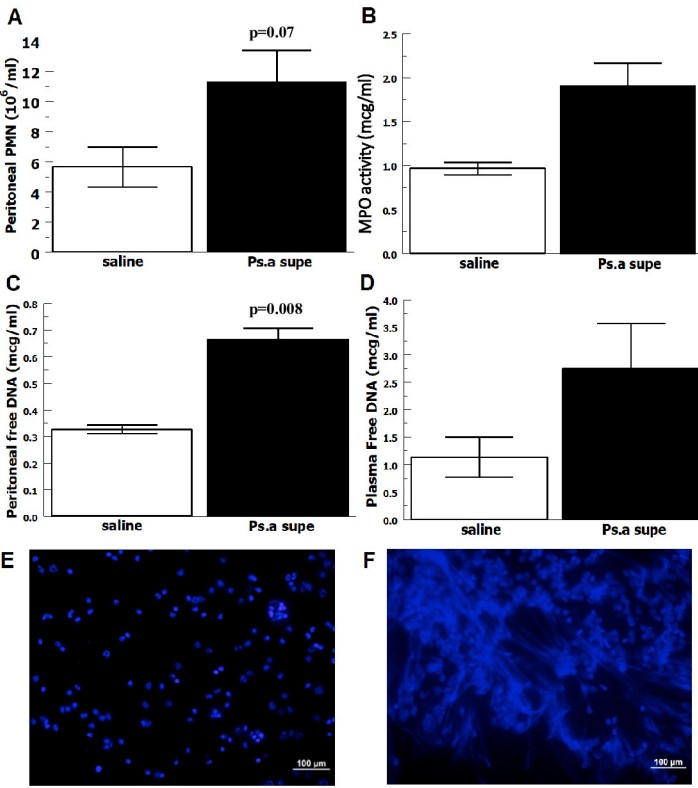

**Fig 2. Neutrophilia, MPO and NETosis in *P. aeruginosa* supernatant (Ps.a supe) induced peritonitis.** Rats received IP injection with thioglycolate to recruit neutrophils and then IP injection with Ps.a supe or saline. After euthanasia, peritoneal lavage was performed, peritoneal fluid recovered and tested as follows. A) Total neutrophil counts by hemocytometer (n = 3). B) Supernatant MPO activity measured by TMB assay (n = 3). C) Supernatant measured for free DNA by PicoGreen assay (n = 3). D) Plasma measured for free DNA by PicoGreen assay (n = 3). Data are means of independent animals ±SEM. E) Saline injection rat peritoneal fluid cytospin DAPI stain shows neutrophils, but no NETs. F) Ps.a supe injection rat peritoneal fluid cytospin DAPI stain shows NETs. Representative images are shown for each treatment group.

that the slides are concentrated cytospins meant to serve only as qualitative representative images and are not intended for quantitative measurements. DAPI stained slides of peritoneal fluid from *P. aeruginosa* supernatant-injected animals demonstrate NETs (Fig 4B), which are not seen in slides from animals receiving saline only (Fig 4A) or *P. aeruginosa* supernatant plus RLS-0071 (Fig 4C and 4D). The microscopic findings are consistent with the free DNA measurement showing that RLS-0071 can inhibit NETosis induced by *P. aeruginosa* supernatant. Supplemental figure (S3 Fig) shows peritoneal fluid cytospun onto a slide and stained with DAPI, for DNA, anti-neutrophil elastase antibodies and anti-histone antibodies. Please note that the slides are concentrated cytospins meant to serve only as qualitative representative images and are not intended for quantitative measurements. Neutrophil elastase and histones are present on DNA webs elaborated as neutrophil-derived NETs. As can been seen in the individually stained images (S3A–S3C Fig) and overlay image (S3D Fig) the DNA webs are decorated with neutrophil elastase and histones consistent with neutrophil-derived NETs. We did not identify extracellular DNA in the peritoneal fluid that was not associated with neutrophil elastase and histones, suggesting that the vast majority of extracellular DNA in the peritoneal fluid is derived from NETs. This is consistent with previously published methodology for

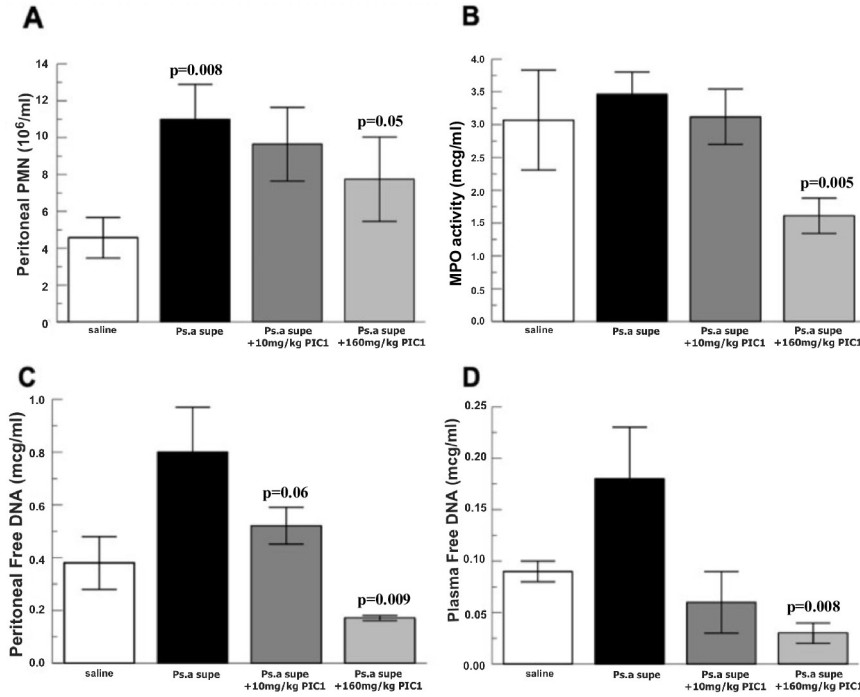

**Fig 3. Peritonitis model of inflammatory response to *P. aeruginosa* broth (minus organisms) shows that RLS-0071 (also known as PIC1) inhibits MPO activity and decreases NETosis.** Rats received IP injection with thioglycolate to recruit neutrophils and then IP injection with *P. aeruginosa* supernatant and doses of RLS-0071 or saline. After euthanasia, peritoneal lavage was performed, and the peritoneal fluid recovered and tested as follows. A) Total neutrophil counts by hemocytometer (n = 4). B) Supernatant MPO activity measured by TMB oxidation (n = 4). C) Supernatant measured for free DNA by PicoGreen assay (n = 4). D) Plasma measured for free DNA by PicoGreen assay (n = 4). Data are means of independent animals ±SEM.

NET quantitation [37,48]. Together these findings suggest that RLS-0071 can inhibit *P. aeruginosa* initiated MPO activity and NETosis in vivo.

## RLS-0071 modulates inflammatory peritonitis induced by immune-complexes

To evaluate RLS-0071's ability to modulate neutrophil release of MPO and inhibit NET formation induced by immune-complexes we adapted the inflammatory peritonitis animal model by injecting preformed immune-complexes IP. Fig 5A shows a trend of 1.3-fold increase of intra-peritoneal neutrophil recruitment induced by immune-complexes (p = 0.5) and Fig 5B shows a significant increase in IP MPO activity induced by immune complexes compared to saline (p = 0.08). Fig 5B shows RLS-0071 at 10 mg/kg decreased MPO activity in the peritoneal fluid by more than 2-fold (p = 0.04). Fig 5C shows a 2-fold increase in free DNA released into the peritoneum induced by immune-complexes compared to saline (p = 0.02) with a > 1.7 fold decrease in NET formation for RLS-0071 at 10 mg/kg (p = 0.06) and 160mg/kg (p = 0.012). High dose RLS-0071 produced a 1.7-fold decreased in free DNA released into the plasma (p = 0.03) when compared to the immune-complexes alone. Fig 6 show representative images of DAPI-stained slides of peritoneal wash supernatants. Please note that the slides are concentrated cytospins meant to serve only as qualitative representative images and are not intended for quantitative measurements. DAPI stained slides of peritoneal fluid from immune complex injected animals demonstrate NETs (Fig 6B), which are not seen in slides from animals

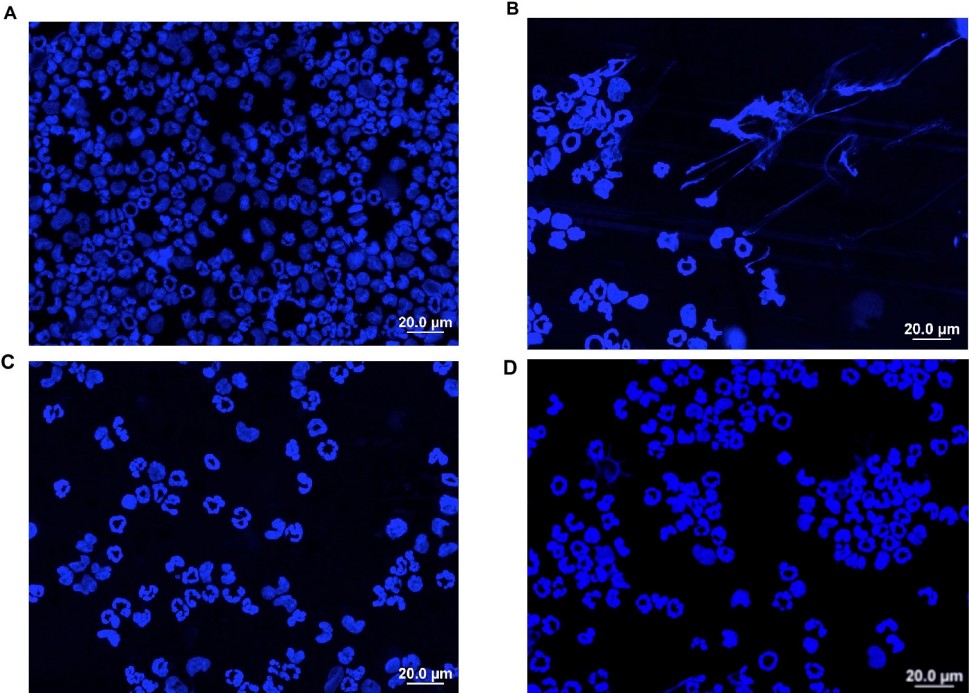

**Fig 4. *P. aeruginosa* supernatant injection induces NETosis visualized by DAPI peritoneal wash slide @60x.** A) Saline injection rat peritoneal fluid cytospin DAPI stain shows neutrophils, but no NETs B) *P. aeruginosa* supernatant injection rat peritoneal fluid cytospin DAPI stain shows NETs. C) RLS-0071at10mg/kg injection rat peritoneal fluid cytospin DAPI stain shows neutrophils, but no NETs. D) RLS-0071 at 160mg/kg injection rat peritoneal fluid cytospin DAPI stain shows neutrophils, but no NETs.

receiving saline only (Fig 6A) or immune complexes plus RLS-0071 (Fig 6C and 6D). These images are consistent with the quantitative free DNA data. Taken together these experiments demonstrate immune-complexes induce inflammatory peritonitis mediated by multiple neutrophil effectors including MPO and NETosis and suggests that RLS-0071 can inhibit these effectors *in vivo*.

## Discussion

Herein we investigated the *in vivo* capability of RLS-0071 to modulate MPO activity and NET formation in a rat model of inflammatory peritonitis. We were able to demonstrate that when purified MPO was injected IP, it was functionally active in oxidizing target substrates which could be successfully inhibited by RLS-0071. Purified MPO in the peritoneal space also produced a dose-dependent effect increasing intraperitoneal free DNA released by neutrophils consistent with MPO being a known trigger of NETosis. Free DNA release by intraperitoneal MPO was demonstrated to be inhibited with increasing doses of RLS-0071. We then induced peritonitis with *P. aeruginosa* supernatant as an inflammatory stimulus that commonly contributes to the disease process in individuals with cystic fibrosis. This supernatant can contain endotoxin, various amounts of proteases [49] and other inflammatory substances to induce a robust inflammatory milieu intraperitoneally. We were able to show that RLS-0071 dose-dependently decreased MPO release and NET formation stimulated by *P. aeruginosa*. Additionally, in a modified reverse passive arthus (RPA) animal model created by injecting pre-formed immune-complexes into the peritoneal cavity, we showed that RLS-0071 was able to similarly reduce MPO activity and NET formation.

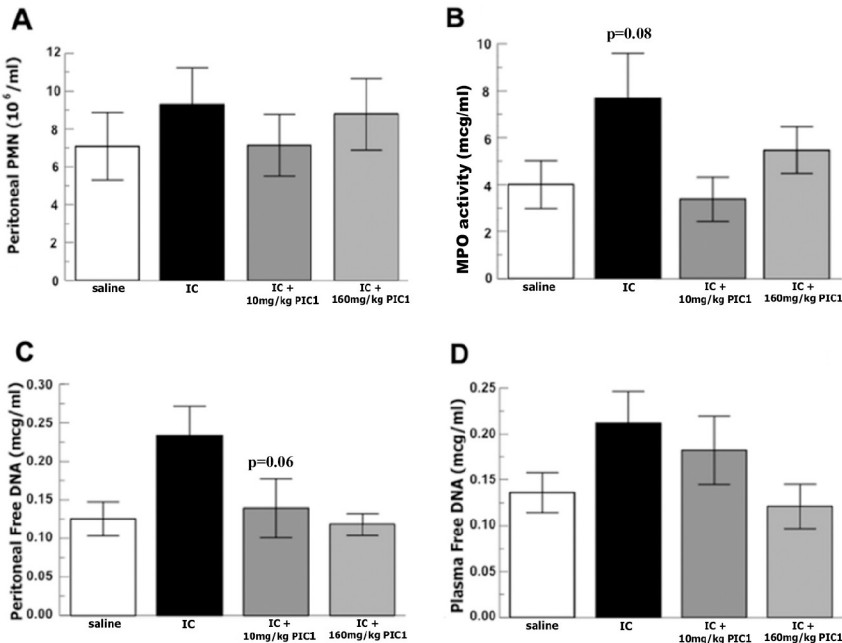

**Fig 5. Peritonitis model of inflammatory response to immune complexes shows that RLS-0071 (also known as PIC1) inhibits MPO activity and decreases NETosis.** Rats received IP injection with thioglycolate to recruit neutrophils and then IP injection with Immune Complexes followed by low vs. high doses of RLS-0071 or saline. After euthanasia, peritoneal lavage was performed, and the peritoneal fluid recovered and tested as follows. A) Total neutrophil counts by hemocytometer (n = 8). B) Supernatant MPO activity measured by TMB oxidation (n = 8). C) Supernatant measured for free DNA by PicoGreen assay (n = 8). D) Plasma measured for free DNA by PicoGreen assay (n = 8). Data are means of independent animals ±SEM.

To our knowledge, this is the first description of a modified RPA reaction utilizing pre-formed immune-complexes in lieu of ovalbumin administered IP followed by intravenous (IV) rabbit anti-ovalbumin sera. The model we describe here uses the peritoneal cavity as a self-contained environment and can be adapted to investigate various inflammatory stimuli and multiple neutrophil responses *in vivo*. The large volumes obtained from a peritoneal wash allows for detailed analysis of a variety of assays. Additionally, this model also allows for evaluation of systemic markers in plasma.

Our current study's findings demonstrate that RLS-0071 can inhibit MPO oxidative activity and dose-dependently reduce NET formation *in vivo* triggered by a variety of potent inflammatory stimuli that contribute to the pathogenesis of cystic fibrosis. This is consistent with our prior findings that RLS-0071 can inhibit MPO oxidative activity utilizing purified MPO *in vitro* [36], in CF sputum samples *ex vivo* [35] and inhibit MPO-triggered NET formation *in vitro* [39]. It has also been shown that complement effectors (C5a, C3a) may significantly impact lung inflammation in CF and CF lung disease is mediated in part, by large influxes of neutrophils into lung tissue elaborating MPO and NETs [16,17,29,30,50]. These experiments provide proof of concept that an agent such as RLS-0071 could potentially moderate these important neutrophil effector functions, that modulate important aspects of CF lung inflammation and may be able to slow the progression of inflammatory lung damage in individuals with cystic fibrosis. In addition, RLS-0071 has been demonstrated to inhibit the growth of pathogenic bacteria such as *P. aeruginosa* [51] that are implicated in neutrophil recruitment in CF and subsequent lung damage. Potential next steps include utilizing an inflammatory

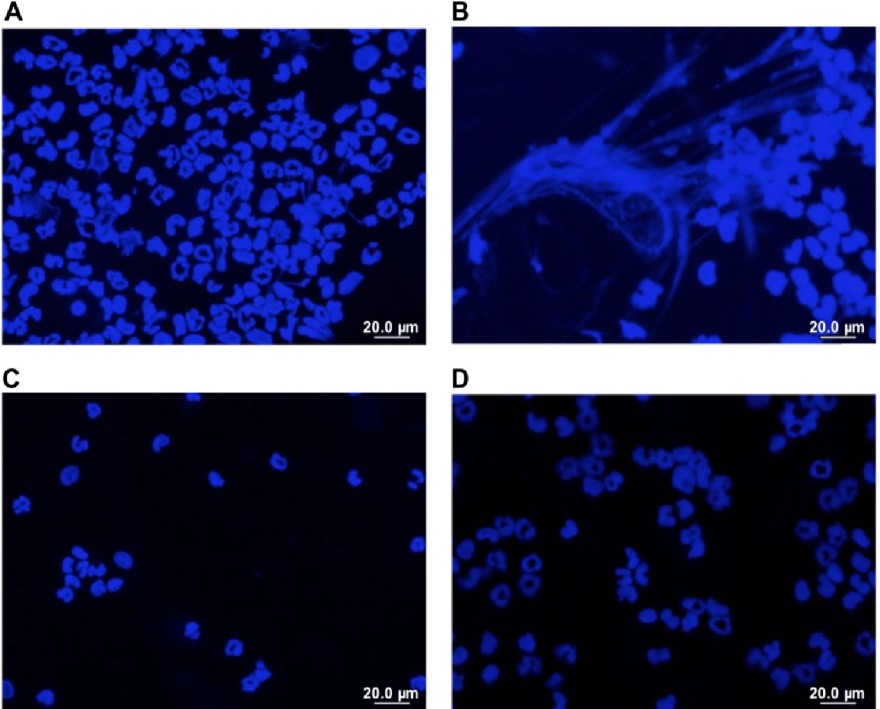

**Fig 6. Immune complex induces NETosis visualized by DAPI peritoneal wash slide @60x.** A) Saline injection rat peritoneal fluid cytospin DAPI stain shows neutrophils, but no NETs B) Immune Complex injection rat peritoneal fluid cytospin DAPI stain shows NETs. C) RLS-0071 at 10mg/kg injection rat peritoneal fluid cytospin DAPI stain shows neutrophils, but no NETs D) RLS-0071 at 160mg/kg injection rat peritoneal fluid cytospin DAPI stain shows neutrophils, but no NETs.

pneumonitis animal model to evaluate the extent to which agents such as RLS-0071 can modify neutrophil effector functions like MPO activity and NETosis in the lung.

## Supporting information

**S1 Fig. Intraperitoneal MPO dose ranging and time course experiments.** Panel A and B show Intraperitoneal MPO dose ranging experiments Increasing amounts of purified MPO was injected IP and after 1 hour animals underwent phlebotomy, euthanasia and peritoneal wash. A) Peritoneal wash supernatant oxidation of TMB (n = 4). Data are means of independent animals ±SEM. B) Peritoneal wash supernatant free DNA measured via PicoGreen assay (n = 4). Data are means of independent animals ±SEM. Panel C and D show Intraperitoneal MPO time course experiments Purified MPO (0.1 mg) was injected IP and at increasing intervals animals underwent phlebotomy, euthanasia and peritoneal wash. C) Peritoneal wash supernatant oxidation of TMB (n = 4). Data are means of independent animals ±SEM. D) Peritoneal wash supernatant free DNA measured via PicoGreen assay (n = 4). Data are means of independent animals ±SEM.
(PDF)

**S2 Fig. Neutrophilia in *P. aeruginosa* supernatant (Ps.a supe) induced peritonitis.** Rats received IP injection with thioglycolate to recruit neutrophils and then IP injection with Ps.a supe or saline. After euthanasia, peritoneal lavage was performed, peritoneal fluid recovered, and Wright stain was performed. A) Saline injection rat peritoneal fluid cytospin stain shows neutrophils B) Ps.a supe injection rat peritoneal fluid cytospin shows neutrophils.

Representative images are shown for each group.
(PDF)

**S3 Fig. *P. aeruginosa* supernatant injection induces NETosis in peritoneal wash slide @60x.** Representative images show Pseudomonas-initiated neutrophil extracellular trap (NET) formation in peritonitis fluid. NET formation assayed by neutrophil elastase (NE) probed with anti-neutrophil elastase (αNE) antibody (A), fluorescence microscopy for DNA (DAPI) (B), histone H3 probed with anti-histone H3 (αhistone) antibody (C). Panel D shows the 3 stains (A,B and C) superimposed demonstrating co-localization confirming that these are NETs. (PDF)

## Author Contributions

**Conceptualization:** Neel Krishna, Kenji Cunnion.

**Data curation:** Adrianne Enos, Brittany Lassiter, Alana Sampson, Pamela Hair.

**Formal analysis:** Parvathi Kumar, Brittany Lassiter, Neel Krishna, Kenji Cunnion.

**Investigation:** Adrianne Enos, Brittany Lassiter, Alana Sampson.

**Methodology:** Adrianne Enos, Brittany Lassiter, Alana Sampson, Pamela Hair, Neel Krishna, Kenji Cunnion.

**Project administration:** Pamela Hair, Neel Krishna.

**Resources:** Pamela Hair, Neel Krishna.

**Software:** Adrianne Enos, Parvathi Kumar, Brittany Lassiter, Pamela Hair.

**Supervision:** Parvathi Kumar, Pamela Hair, Neel Krishna, Kenji Cunnion.

**Writing – original draft:** Parvathi Kumar.

**Writing – review & editing:** Adrianne Enos, Parvathi Kumar, Brittany Lassiter, Alana Sampson, Pamela Hair, Neel Krishna, Kenji Cunnion.

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
