## [Decision Letter · Decision Letter 0]

6 Apr 2021

PONE-D-20-35764

Peptide inhibition of myeloperoxidase activity and neutrophil extracellular traps after in vivo challenge with Pseudomonas aeruginosa supernatant and immune-complexes

PLOS ONE

Dear Dr. Kumar,

Thank you for submitting your manuscript to PLOS ONE. After careful consideration, we feel that it has merit but does not fully meet PLOS ONE’s publication criteria as it currently stands. Therefore, we invite you to submit a revised version of the manuscript that addresses the points raised during the review process.

Your manuscript was reviewed by two experts and we received positive feedback. However there are questions remained unanswered.

We look forward to receiving your revised manuscript.

Kind regards,

Partha Mukhopadhyay, Ph.D.

Academic Editor

PLOS ONE

Journal Requirements:

2. As part of your revisions, please provide details about humane endpoints for the animals, and also indicate whether there were any unanticipated adverse events related to the animal study. We thank you for your attention in this matter.

[The author(s) received no specific funding for this work.].   

We note that one or more of the authors are employed by a commercial company: ReAlta Life Sciences Inc.

Reviewers' comments:

Reviewer's Responses to Questions

**Comments to the Author**

1. Is the manuscript technically sound, and do the data support the conclusions?

Reviewer #1: Yes

Reviewer #2: Partly

2. Has the statistical analysis been performed appropriately and rigorously? 

Reviewer #1: Yes

Reviewer #2: No

3. Have the authors made all data underlying the findings in their manuscript fully available?

Reviewer #1: Yes

Reviewer #2: Yes

4. Is the manuscript presented in an intelligible fashion and written in standard English?

Reviewer #1: Yes

Reviewer #2: Yes

5. Review Comments to the Author

Reviewer #1: In the present manuscript Kumar et al. investigated the capability of a 15 amino acid PEGylated synthetic peptide, also known as Peptide Inhibitor of Complement C1 or RLS-0071, to modulate neutrophil MPO activity and NET formation in vivo in a rat inflammatory peritonitis model. The authors showed that PIC1 inhibited MPO activity and NETosis, reduced P. aeruginosa supernatant induced neutrophil infiltration, as well as inhibited the 2 monitored neutrophil effector functions in a modified reverse passive Arthus model induced by intraperitoneal injection of preformed immune-complexes.

Although the goals are clear and the presented data support findings of the manuscripts, there are some issues that needs to be addressed or corrected in the final version:

1. Figure 1. shows huge SEMs. Can authors improve this figure by including more animal to the study or put it to supplementary figure?

2. Peroxidases need H2O2 in order to oxidize any natural (Cl-, Br-, I-, SCN-, Tyrosine etc.) or artificial (TMB, O-dianisidine) substrate. In the Methods section of Quantitation of MPO peroxidase activity the authors use TMB substrate, but they don`t mention H2O2, if they added to the TMB reaction or used a commercial TMB solution which included. This question is particularly important when authors determine peroxidase activity in large volumes as the 20 mL peritoneal wash fluid. Please address this question and provide reaction equation, TMB solution concentration, H2O2 content or manufacturer.

3. In the discussion authors mention cystic fibrosis disease, where among other factors and pathogens P. aeruginosa induced neutrophil recruitment, NET formation and MPO release would contribute to lung tissue damage, on the long-term respiratory failure. Please discuss potential clinical use of RLS-0071 in treatment of cystic fibrosis. How could it be administered (IV, oral, aerosol etc.)? Do the authors have any data regarding the pharmacokinetics of RLS-0071 (e.g.: tissue distribution, half-life)? Since it’s a 15 aa peptide fragment, will it evoke immune reaction? What is PEGylation for?

4. Figure 4A: PIC1 IP administration reduced P. aeruginosa induced PMN recruitment. What is the mechanism?

5. Figure 5 A and C: 5A DAPI staining belongs to saline injection of Figure 4A with about 5x106/ml PMN infiltration. Figure 5C representative image shows much less cells even though from Figure 4A Ps.a. sup + 10 mg/kg PIC1 shows app. 10x106/ml PMN. Please address this discrepancy or use a different representative picture.

6. Same PMN number issue on Figure 7A and 7C, while the belonging infiltrated PMN count is comparable on Figure 6A. Why is this discrepancy?

Minor comments, typos:

1. MPO has peroxidase or oxidase activity and not “oxidant or oxidative activity” like chemicals or compounds. Line: 187, 237, 344

2. On Figures 1, 2, 3, 4, 6 statistical significance should be marked when applicable.

3. On Figures 1A, 1C, 2A, 3B, 4B, 6B “MPO oxidize TMB” Y axis label should be corrected to MPO activity or oxidized TMB etc.

4. Line 250: missing p value at 4 hours MPO treatment.

5. Line 270: “peritoneal fluid was performed after euthanasia” peritoneal fluid can only be collected or extracted not performed.

6. Line 331: decreased MPO release and not “decreased MPO released”

Reviewer #2: In the study titled, 'Peptide inhibition of myeloperoxidase activity and neutrophil extracellular traps after in vivo challenge with Pseudomonas aeruginosa supernatant and immune-complexes', the authors have attempted to demonstrate a role for peptide RLS-0071 in inhibiting myeloperoxidase activity and neutrophil extracellular traps relevant in the context of inflammatory and autoimmune diseases such as cystic fibrosis (CF). There were numerous issues with this study which are listed below:

1) The title of the study does not really tell you anything. It is suggested the authors go with some iteration of the shorter running title.

2) The abstract needs to be rewritten. The authors jump to myeloperoxidase, NETs and then to a brief mention of inflammatory diseases in a haphazard manner without giving much context or clarifying rationale.

3) Introduction also needs a lot of work. Give more background on all points especially on cystic fibrosis and it's pathophysiologies and related immune response. More background on myeloperoxidase and NETS and their relation to CF. CF and psedumonas aeruginosa etc.

4) The use of the peritoneal cavity as a model system is questionable. Yes, CF patients do show GI pathophysiology, but the type of immune cells associated with this include B cells and macrophages and not neutrophils, so not sure how the authors are using this to measure the MPO activity that is associated with neutrophils specifically. The peritoneal cavity is better suited to other inflammatory and autoimmune conditions like colitis, Crohn's, IBS, lupus etc. There are better established in vivo models to study effects of CF and effects of drugs to treat CF as highlighted by a study published by Rosen B.H. et. al. in the Journal of Cystic Fibrosis (2018). It is suggested that the authors look into these.

5) Need more detail on the principle behind the quantitation of NET formation assay.

6) How can it be assumed that the free DNA found circulating in the peritoneal cavity is from neutrophils?

7) In the experiments showing NET formation as measured by immunofluorescence microscopy, how can the authors prove that the cells that we are looking at are indeed neutrophils? They should repeat these experiments using by staining for neutrophil markers like CD66b/CEACAM-8, CD11b/Integrin alpha M, CD33 etc. Also, stain for myeloperoxidase with and without RLS-0071.

8) The authors have not indicated how many mice were used in each of the experiments. Is n=4, the number of mice, or experimental repeats? If n=4 is the number of mice in each experimental group, then it is not a big enough sample size to get statistically significant results. Power analysis needs to be done, and the authors should do non-parametric tests. Also, please redo statistical analysis using a software like Graphpad prism.

9)RLS-0071 affects oxidative abilities of hemoglobin and myoglobin which are essential cellular proteins. This might lead to adverse side effects in the body. How do the authors reconcile this?

10) Showing inhibition of myeloperoxidase activity and NETs in itself is not sufficient to show that RLS-0071 will be beneficial in treating CF as a which is disease with multiple pathophysiologies. Overall, the authors are not clear exactly what clinical disease the authors are attempting to alleviate with RLS-0071. They start of with CF, but they end their discussion saying this peptide can be used to treat SLE. Clinical significance is unclear.

11) Authors say, 'Potential next steps include utilizing an inflammatory pneumonitis animal model to evaluate the extent to which agents such as RLS-0071 can modify neutrophil effector functions like MPO activity and NETosis in the lung. Kindly complete this part and resubmit this manuscript.

6. PLOS authors have the option to publish the peer review history of their article (what does this mean?). If published, this will include your full peer review and any attached files.

Reviewer #1: No

Reviewer #2: No

---

## [Author Response · Author response to Decision Letter 0]

18 May 2021

Reviewers' comments:

Reviewer 1 comments to Author: 

General comments: 

In the present manuscript Kumar et al. investigated the capability of a 15 amino acid PEGylated synthetic peptide, also known as Peptide Inhibitor of Complement C1 or RLS-0071, to modulate neutrophil MPO activity and NET formation in vivo in a rat inflammatory peritonitis model. The authors showed that PIC1 inhibited MPO activity and NETosis, reduced P. aeruginosa supernatant induced neutrophil infiltration, as well as inhibited the 2 monitored neutrophil effector functions in a modified reverse passive Arthus model induced by intraperitoneal injection of preformed immune-complexes.

Although the goals are clear and the presented data support findings of the manuscripts, there are some issues that needs to be addressed or corrected in the final version:

1. Figure 1. shows huge SEMs. Can authors improve this figure by including more animal to the study or put it to supplementary figure?

Author’s response: We appreciate the input. The experiment was performed as a pilot study to inform MPO dose and timing for the follow up experiments. We have elected to convert the figure to a supplementary figure.

2. Peroxidases need H2O2 in order to oxidize any natural (Cl-, Br-, I-, SCN-, Tyrosine etc.) or artificial (TMB, O-dianisidine) substrate. In the Methods section of Quantitation of MPO peroxidase activity the authors use TMB substrate, but they don`t mention H2O2, if they added to the TMB reaction or used a commercial TMB solution which included. This question is particularly important when authors determine peroxidase activity in large volumes as the 20 mL peritoneal wash fluid. Please address this question and provide reaction equation, TMB solution concentration, H2O2 content or manufacturer.

Author’s response: We appreciate the concern raised here and have clarified the source of TMB as commercial ready to use TMB which includes all the components of the reaction , including hydrogen peroxidase, necessary for detection of MPO peroxidase activity. 

3. In the discussion authors mention cystic fibrosis disease, where among other factors and pathogens P. aeruginosa induced neutrophil recruitment, NET formation and MPO release would contribute to lung tissue damage, on the long-term respiratory failure. Please discuss potential clinical use of RLS-0071 in treatment of cystic fibrosis. How could it be administered (IV, oral, aerosol etc.)? Do the authors have any data regarding the pharmacokinetics of RLS-0071 (e.g.: tissue distribution, half-life)? Since it’s a 15 aa peptide fragment, will it evoke immune reaction? What is PEGylation for?

Author’s response: As part of extensive rational drug design RLS-0071, also known as PIC1, was derived from the human Astrovirus coat protein (1, 2). PEGylation was performed to increase the solubility of the peptide (3). In pre-clinical studies RLS-0071 was shown to be rapidly distributed after I.V infusion from the central compartment to the peripheral compartment (unpublished data). RLS-0071 penetrated into all tissues including lung and was still detectable in the tissues 48 hours after a single dose I.V. (unpublished data). 

RLS-0071 is currently in a healthy volunteer human trail evaluating safety, tolerability and pharmacokinetics after I.V. administration under a Health Canada approved protocol (HC6-24-c247592). This trial is in progress and information will be made available after the clinical study report is finalized.

It has been shown that complement effectors ( C5a, C3a) may significantly impact lung inflammation in CF and CF lung disease is mediated in part by large influxes of neutrophils into lung tissue and elaborating MPO and NETs(4-6) (7, 8). Our prior findings show that RLS-0071 can inhibit MPO oxidative activity in CF sputum samples ex vivo (9) and inhibit NET formation in vitro (2). Compounds, such as RLS-0071, that modulate important aspects of CF lung inflammation including complement and neutrophil effectors may be able to slow the progression of inflammatory lung damage in individuals with cystic fibrosis. 

4. Figure 4A: PIC1 IP administration reduced P. aeruginosa induced PMN recruitment. What is the mechanism?

Author’s response: The early recruitment of PMN to sites of infection is largely driven by generation of complement activation derived chemoattractants and anaphylatoxins C5a and C3a. Inhibition of the classical and lectin complement pathways with RLS-0071 reduces the generation of complement chemoattractants and neutrophil recruitment in response to inflammatory P. aeruginosa supernatant. 

5. Figure 5 A and C: 5A DAPI staining belongs to saline injection of Figure 4A with about 5x106/ml PMN infiltration. Figure 5C representative image shows much less cells even though from Figure 4A Ps.a. sup + 10 mg/kg PIC1 shows app. 10x106/ml PMN. Please address this discrepancy or use a different representative picture.

Author’s response: The slides are concentrated cytospins meant to serve only as qualitative representative images and are not intended for quantitative measurements. This clarification has been added to the results section.

6. Same PMN number issue on Figure 7A and 7C, while the belonging infiltrated PMN count is comparable on Figure 6A. Why is this discrepancy?

Author’s response: The slides are concentrated cytospins meant to serve only as qualitative representative images and are not intended for quantitative measurements. This clarification has been added to the results section.

Minor comments, typos:

1. MPO has peroxidase or oxidase activity and not “oxidant or oxidative activity” like chemicals or compounds. Line: 187, 237, 344

2. On Figures 1, 2, 3, 4, 6 statistical significance should be marked when applicable.

3. On Figures 1A, 1C, 2A, 3B, 4B, 6B “MPO oxidize TMB” Y axis label should be corrected to MPO activity or oxidized TMB etc.

4. Line 250: missing p value at 4 hours MPO treatment.

5. Line 270: “peritoneal fluid was performed after euthanasia” peritoneal fluid can only be collected or extracted not performed.

6. Line 331: decreased MPO release and not “decreased MPO released”

Author’s response: Thank you for these suggestions. We have made all the edits suggested.

Reviewer 2: 

General comments. In the study titled, 'Peptide inhibition of myeloperoxidase activity and neutrophil extracellular traps after in vivo challenge with Pseudomonas aeruginosa supernatant and immune-complexes', the authors have attempted to demonstrate a role for peptide RLS-0071 in inhibiting myeloperoxidase activity and neutrophil extracellular traps relevant in the context of inflammatory and autoimmune diseases such as cystic fibrosis (CF). 

There were numerous issues with this study which are listed below:

1) The title of the study does not really tell you anything. It is suggested the authors go with some iteration of the shorter running title.

Author’s response: Thank you for the feedback and suggestion. We have modified the title. It now reads “ Peptide inhibition of neutrophil-mediated injury after in vivo challenge with supernatant of Pseudomonas aeruginosa and immune-complexes”

2) The abstract needs to be rewritten. The authors jump to myeloperoxidase, NETs and then to a brief mention of inflammatory diseases in a haphazard manner without giving much context or clarifying rationale.

Authors response: Thank you for the feedback. We have edited the abstract to increase context and rationale.

3) Introduction also needs a lot of work. Give more background on all points especially on cystic fibrosis and it's pathophysiologies and related immune response. More background on myeloperoxidase and NETS and their relation to CF. CF and psedumonas aeruginosa etc.

Authors response: Thank you for the feedback. We have edited the introduction to increase the background information on MPO, NETs, P.aeruginosa and CF.

4) The use of the peritoneal cavity as a model system is questionable. Yes, CF patients do show GI pathophysiology, but the type of immune cells associated with this include B cells and macrophages and not neutrophils, so not sure how the authors are using this to measure the MPO activity that is associated with neutrophils specifically. The peritoneal cavity is better suited to other inflammatory and autoimmune conditions like colitis, Crohn's, IBS, lupus etc. There are better established in vivo models to study effects of CF and effects of drugs to treat CF as highlighted by a study published by Rosen B.H. et. al. in the Journal of Cystic Fibrosis (2018). It is suggested that the authors look into these.

Authors response: We agree that there are better-established animal models to study CF. We adapted an established murine peritonitis model for evaluating MPO peroxidase activity and inflammation for 3 reasons:

(1) Rat models of peritonitis are well established for elucidating inflammatory responses, particularly complement system and neutrophil responses, yielding highly interpretable data with minimal sample manipulation (10-13).

(2) Peritoneal lavage provides large volumes of fluid from each rat that can be run in multiple assays (14) reducing the numbers of animals needed and allowing for paired statistical analysis.

(3) We have successfully dosed over 200 rats with RLS-0071 demonstrating efficacy for achieving complement inhibition (15) without evidence of toxicity. 

We chose the inflammatory peritonitis animal model to study neutrophil effects on inflammatory stimuli including bacteria and immune complexes. These stimuli are known mediators of CF lung disease and the intent was to be able to draw parallels between effects seen in an inflammatory milieu and the potential for therapeutic agents targeting these interlinked pathways.

We agree with the reviewer that evaluation in an animal lung model of CF is an important future direction.

5) Need more detail on the principle behind the quantitation of NET formation assay.

Authors response: Neutrophil extracellular traps (NETs) are released by neutrophils in a web of decondensed chromatin fibres decorated with antimicrobial factors delivered by the granules. The most important feature specific to NETs is the presence of DNA fibers in the extracellular space (16) which can be quantified using PicoGreen (Invitrogen) , a fluorochrome that selectively binds dsDNA and is the most common method to quantify circulating cell-free DNA/NETs (17-19). 

6) How can it be assumed that the free DNA found circulating in the peritoneal cavity is from neutrophils?

Authors response: Please see response to question 5. 

In addition we have provided a supplemental figure (S3 Fig) which shows peritoneal fluid cytospun onto a slide and stained with DAPI, for DNA, anti-neutrophil elastase antibodies and anti-histone antibodies. Neutrophil elastase and histones are present on DNA webs elaborated as neutrophil-derived NETs. As can been seen in the individually stained images (S3 A, S3 B, S3 C) and overlay image (S3 D) the DNA webs are decorated with neutrophil elastase and histones consistent with neutrophil-derived NETs. We did not identify extracellular DNA in the peritoneal fluid that was not associated with neutrophil elastase and histones, suggesting that the vast majority of extracellular DNA in the peritoneal fluid is derived from NETs. This is consistent with previously published methodology for NET quantitation (20). 

The preceding description has been added to the results section.

7) In the experiments showing NET formation as measured by immunofluorescence microscopy, how can the authors prove that the cells that we are looking at are indeed neutrophils? They should repeat these experiments using by staining for neutrophil markers like CD66b/CEACAM-8, CD11b/Integrin alpha M, CD33 etc. Also, stain for myeloperoxidase with and without RLS-0071.

Authors response: The DAPI stained images demonstrate that the vast majority of the visualized cells have a multi-lobed nucleus, which are only seen for neutrophils. Wright stains were also performed on the cytospun peritoneal fluid confirming the cells to be neutrophils (S2 Fig). 

In our prior publications, we have shown that RLS-0071 (also known as PIC1) can inhibit PMA-stimulated NET formation by human neutrophils (1, 2) by demonstrating co-localization of the major components of NETs , extracellular DNA , myeloperoxidase, neutrophil elastase and histones. 

For this submission, a supplemental figure has been included wherein we show representative images of P. aeruginosa supernatant injection induced NETosis. The cytospins were stained for neutrophil elastase and histone to ascertain neutrophils and stained for DNA using DAPI. The superimposed figure (S3 Fig panel D) confirms that the DNA webs are NETs.

8) The authors have not indicated how many mice were used in each of the experiments. Is n=4, the number of mice, or experimental repeats? If n=4 is the number of mice in each experimental group, then it is not a big enough sample size to get statistically significant results. Power analysis needs to be done, and the authors should do non-parametric tests. Also, please redo statistical analysis using a software like Graphpad prism.

Authors response: We appreciate the reviewer’s concern about our statistical approach. Please note that all of these experiments were performed in Wistar rats and not mice. N = 4 is the number of rats in each experimental cohort. Utilizing much larger animals allows us to conduct these in vivo experiments using much larger volumes of peritoneal wash fluid, 20 ml of ice-cold saline, and recover similarly large volumes of fluid. Blood draws were via tail vein phlebotomy and yielded 0.5 – 1 ml of blood. Thus, all of these assays were performed with large volumes of sample decreasing the animal-to-animal variability typically seen for mice. We have previously published data utilizing the same techniques in a Wistar rat model of S. aureus peritonitis which yielded highly interpretable data. (14)

In the analysis of human data an n = 4 would be inadequate due to the high degree of variability of genetic background, age, diet, lifestyle, etc. However, in experiments utilizing rats of a single strain, housed under the same conditions, eating the same diet and undergoing identical procedures, the variability is highly constrained, and this is reflected in the tight error bars seen for these data. Non-parametric tests are warranted for small sample sizes where there is reason to doubt that the data conforms with a normal distribution. Our data show no evidence of skew and we have no reason to suspect that the data is not normally distributed. This is typically the case for these types of tightly controlled animal experiments and for this reason student’s t-test remains the standard method for analyzing these types of data. There is also the ethical consideration against conducting animal experiments with large numbers of animals when much smaller numbers (e.g. n = 4) can yield meaningful results. Please see the following reference detailing the argument for utilizing the minimal numbers of animals possible and t-tests in animal experimentation (21) (Michael F. W. Festing, Douglas G. Altman, Guidelines for the Design and Statistical Analysis of Experiments Using Laboratory Animals, ILAR Journal, Volume 43, Issue 4, 2002, Pages 244–258, https://doi.org/10.1093/ilar.43.4.244).

9)RLS-0071 affects oxidative abilities of hemoglobin and myoglobin which are essential cellular proteins. This might lead to adverse side effects in the body. How do the authors reconcile this?

Authors response: RLS-0071 exerts reversible antioxidant activity when these proteins are extracellular eg: peroxidase activity of MPO in sputum acquired from individuals with CF 

(9) and in the peroxidase activity of extracellular Hb ( RBC lysates) (22). RLS-0071 does not penetrate intracellularly, thus there is no concern for affecting the function of erythrocytes or myocytes. However, extracellular hemoglobin and myoglobin are toxic to kidneys and in sufficient quantities will precipitate acute kidney injury. We have previously shown in an animal model of hemolysis that RLS-0071 can protect the kidneys from damage (15) .

10) Showing inhibition of myeloperoxidase activity and NETs in itself is not sufficient to show that RLS-0071 will be beneficial in treating CF as a which is disease with multiple pathophysiologies. Overall, the authors are not clear exactly what clinical disease the authors are attempting to alleviate with RLS-0071. They start of with CF, but they end their discussion saying this peptide can be used to treat SLE. Clinical significance is unclear.

Authors response: Thank you for the feedback. We have focused clinical implications for CF in the introduction and discussion. We agree with the reviewer that CF is a complex and multifactorial disease process. We believe what is novel here is targeting aspects of inflammatory lung damage in CF that currently have no therapeutic options for modulation or treatment.

11) Authors say, 'Potential next steps include utilizing an inflammatory pneumonitis animal model to evaluate the extent to which agents such as RLS-0071 can modify neutrophil effector functions like MPO activity and NETosis in the lung. Kindly complete this part and resubmit this manuscript.

Authors response: We have completed this as a separate set of experiments in an animal model of acute lung injury model and submitted an entire separate manuscript which is currently under review with PLOS One. However, these animal models are completely different and deserve separate manuscripts. This manuscript focuses on the inflammatory mechanisms of action, which are best suited for evaluation in this peritonitis model. The acute lung injury model focuses on evaluating multiple aspects of lung damage including assaying lung histology and bronchoalveolar lavage fluid.

1. Hair PS, Enos AI, Krishna NK, Cunnion KM. Inhibition of complement activation, myeloperoxidase, NET formation and oxidant activity by PIC1 peptide variants. PLoS One. 2019;14(12):e0226875.

2. Hair PS, Enos AI, Krishna NK, Cunnion KM. Inhibition of Immune Complex Complement Activation and Neutrophil Extracellular Trap Formation by Peptide Inhibitor of Complement C1. Front Immunol. 2018;9:558.

3. Sharp JA, Hair PS, Pallera HK, Kumar PS, Mauriello CT, Nyalwidhe JO, et al. Peptide Inhibitor of Complement C1 (PIC1) Rapidly Inhibits Complement Activation after Intravascular Injection in Rats. PLoS One. 2015;10(7):e0132446.

4. Thomson E, Brennan S, Senthilmohan R, Gangell CL, Chapman AL, Sly PD, et al. Identifying peroxidases and their oxidants in the early pathology of cystic fibrosis. Free Radic Biol Med. 2010;49(9):1354-60.

5. Hair PS, Sass LA, Vazifedan T, Shah TA, Krishna NK, Cunnion KM. Complement effectors, C5a and C3a, in cystic fibrosis lung fluid correlate with disease severity. PLoS One. 2017;12(3):e0173257.

6. Sass LA, Hair PS, Perkins AM, Shah TA, Krishna NK, Cunnion KM. Complement Effectors of Inflammation in Cystic Fibrosis Lung Fluid Correlate with Clinical Measures of Disease. PLoS One. 2015;10(12):e0144723.

7. Witko-Sarsat V, Delacourt C, Rabier D, Bardet J, Nguyen AT, Descamps-Latscha B. Neutrophil-derived long-lived oxidants in cystic fibrosis sputum. Am J Respir Crit Care Med. 1995;152(6 Pt 1):1910-6.

8. Gray RD, Hardisty G, Regan KH, Smith M, Robb CT, Duffin R, et al. Delayed neutrophil apoptosis enhances NET formation in cystic fibrosis. Thorax. 2018;73(2):134-44.

9. Hair PS, Sass LA, Krishna NK, Cunnion KM. Inhibition of Myeloperoxidase Activity in Cystic Fibrosis Sputum by Peptide Inhibitor of Complement C1 (PIC1). PLoS One. 2017;12(1):e0170203.

10. Bestebroer J, Aerts PC, Rooijakkers SH, Pandey MK, Kohl J, van Strijp JA, et al. Functional basis for complement evasion by staphylococcal superantigen-like 7. Cellular microbiology. 2010;12(10):1506-16.

11. Steil AA, Garcia Rodriguez MC, Alonso A, Crespo MS, Bosca L. Platelet-activating factor: the effector of protein-rich plasma extravasation and nitric oxide synthase induction in rat immune complex peritonitis. British journal of pharmacology. 1995;114(4):895-901.

12. Strachan AJ, Woodruff TM, Haaima G, Fairlie DP, Taylor SM. A new small molecule C5a receptor antagonist inhibits the reverse-passive Arthus reaction and endotoxic shock in rats. Journal of immunology (Baltimore, Md : 1950). 2000;164(12):6560-5.

13. Bestebroer J, Aerts PC, Rooijakkers SH, Pandey MK, Köhl J, Van Strijp JA, et al. Functional basis for complement evasion by staphylococcal superantigen‐like 7. Cellular microbiology. 2010;12(10):1506-16.

14. Mauriello CT, Hair PS, Rohn RD, Rister NS, Krishna NK, Cunnion KM. Hyperglycemia inhibits complement-mediated immunological control of S. aureus in a rat model of peritonitis. J Diabetes Res. 2014;2014:762051.

15. Kumar PS, Pallera HK, Hair PS, Rivera MG, Shah TA, Werner AL, et al. Peptide inhibitor of complement C1 modulates acute intravascular hemolysis of mismatched red blood cells in rats. Transfusion. 2016;56(8):2133-45.

16. Brinkmann V, Reichard U, Goosmann C, Fauler B, Uhlemann Y, Weiss DS, et al. Neutrophil extracellular traps kill bacteria. Science. 2004;303(5663):1532-5.

17. Meng W, Paunel-Görgülü A, Flohé S, Hoffmann A, Witte I, MacKenzie C, et al. Depletion of neutrophil extracellular traps in vivo results in hypersusceptibility to polymicrobial sepsis in mice. Critical care. 2012;16(4):1-13.

18. Saffarzadeh M, Juenemann C, Queisser MA, Lochnit G, Barreto G, Galuska SP, et al. Neutrophil extracellular traps directly induce epithelial and endothelial cell death: a predominant role of histones. PLoS One. 2012;7(2):e32366.

19. Akong-Moore K, Chow OA, von Köckritz-Blickwede M, Nizet V. Influences of chloride and hypochlorite on neutrophil extracellular trap formation. PloS one. 2012;7(8):e42984.

20. Fuchs TA, Abed U, Goosmann C, Hurwitz R, Schulze I, Wahn V, et al. Novel cell death program leads to neutrophil extracellular traps. Journal of Cell Biology. 2007;176(2):231-41.

21. Festing MF, Altman DG. Guidelines for the design and statistical analysis of experiments using laboratory animals. ILAR journal. 2002;43(4):244-58.

22. Hair PS, Cunnion KM, Krishna NK. Peptide Inhibitor of Complement C1 Inhibits the Peroxidase Activity of Hemoglobin and Myoglobin. Int J Pept. 2017;2017:9454583.

---

## [Decision Letter · Decision Letter 1]

25 Jun 2021

Peptide inhibition of neutrophil-mediated injury after in vivo challenge with supernatant of Pseudomonas aeruginosa and immune-complexes

PONE-D-20-35764R1

Dear Dr. Kumar,

We’re pleased to inform you that your manuscript has been judged scientifically suitable for publication and will be formally accepted for publication once it meets all outstanding technical requirements.

Kind regards,

Partha Mukhopadhyay, Ph.D.

Section Editor

PLOS ONE

Additional Editor Comments (optional):

Reviewers' comments:

Reviewer's Responses to Questions

**Comments to the Author**

1. If the authors have adequately addressed your comments raised in a previous round of review and you feel that this manuscript is now acceptable for publication, you may indicate that here to bypass the “Comments to the Author” section, enter your conflict of interest statement in the “Confidential to Editor” section, and submit your "Accept" recommendation.

Reviewer #1: All comments have been addressed

Reviewer #2: All comments have been addressed

2. Is the manuscript technically sound, and do the data support the conclusions?

Reviewer #1: Yes

Reviewer #2: Yes

3. Has the statistical analysis been performed appropriately and rigorously? 

Reviewer #1: Yes

Reviewer #2: Yes

4. Have the authors made all data underlying the findings in their manuscript fully available?

Reviewer #1: Yes

Reviewer #2: Yes

5. Is the manuscript presented in an intelligible fashion and written in standard English?

Reviewer #1: Yes

Reviewer #2: Yes

6. Review Comments to the Author

Reviewer #1: The authors have correctly answered each of my questions. I accept their response and have no further comments. Congratulations!

Reviewer #2: (No Response)

7. PLOS authors have the option to publish the peer review history of their article (what does this mean?). If published, this will include your full peer review and any attached files.

Reviewer #1: No

Reviewer #2: No

---

## [Editor Report · Acceptance letter]

30 Jun 2021

PONE-D-20-35764R1 

Peptide inhibition of neutrophil-mediated injury after *in vivo* challenge with supernatant of Pseudomonas aeruginosa and immune-complexes 

Dear Dr. Kumar:

I'm pleased to inform you that your manuscript has been deemed suitable for publication in PLOS ONE. Congratulations! Your manuscript is now with our production department. 

Kind regards, 

on behalf of

Dr. Partha Mukhopadhyay 

Section Editor

PLOS ONE